# Simulating IoT Workflows in DISSECT-CF-Fog

**DOI:** 10.3390/s23031294

**Published:** 2023-01-23

**Authors:** Andras Markus, Ali Al-Haboobi, Gabor Kecskemeti, Attila Kertesz

**Affiliations:** 1Department of Software Engineering, University of Szeged, 6720 Szeged, Hungary; 2Institute of Information Technology, University of Miskolc, 3515 Miskolc, Hungary; 3Faculty of Computer Science and Mathematics, University of Kufa, Najaf 54001, Iraq

**Keywords:** Internet of Things, workflow management, fog computing, simulation

## Abstract

The modelling of IoT applications utilising the resources of cloud and fog computing is not straightforward because they have to support various trigger-based events that make human life easier. The sequence of tasks, such as performing a service call, receiving a data packet in the form of a message sent by an IoT device, and managing actuators or executing a computational task on a virtual machine, are often associated with and composed of IoT workflows. The development and deployment of such IoT workflows and their management systems in real life, including communication and network operations, can be complicated due to high operation costs and access limitations. Therefore, simulation solutions are often applied for such purposes. In this paper, we introduce a novel simulator extension of the DISSECT-CF-Fog simulator that leverages the workflow scheduling and its execution capabilities to model real-life IoT use cases. We also show that state-of-the-art simulators typically omit the IoT factor in the case of the scientific workflow evaluation. Therefore, we present a scalability study focusing on scientific workflows and on the interoperability of scientific and IoT workflows in DISSECT-CF-Fog.

## 1. Introduction

With increasing mobility and connectivity, the Internet of Things (IoT) has become one of the most important paradigms in information technology in the 21st century [1]. In the era of the IoT, various devices and their sensors produce a huge number of datasets, and data processing solutions consist of a set of activities or tasks that can be executed in the cloud or at the edge of the network [2]. Cloud computing is often used by IoT systems, because it is ubiquitous and offers theoretically unlimited elastic computing and storage services, which fit the needs of IoT applications.

Due to the growing number of IoT applications and their time-critical requirements, purely cloud-based solutions are insufficient, and delays often occur because cloud data centres are usually far away from the data sources. Fog computing was derived from cloud computing to solve the problems of increasing latency, the high density of smart devices, and congested communication channels, also known as the bottleneck effect [3]. To overcome the problems associated with centralised management, it is crucial for time-critical applications to use fog resources and only benefit from the cloud when needed. However, there are a number of obstacles to consider, such as maximising the use of fog resources while taking into account their computational limitations. In addition, some of the tasks may be time-critical, requiring immediate allocation and execution. The execution of upcoming tasks requires effective scheduling and task allocation that meets all requirements.

Since modern applications are no longer built as monolithic architectures but are composed from small software components (e.g., microservices) with dependencies, such applications can be described as workflows. Due to the unique nature of the IoT, workflows incorporating IoT activities must adapt to these new environments. Data are generated in huge quantities because of the large number of sensors, smart devices, and gadgets in use today. Therefore, in the context of the IoT, fog computing is often used to improve data processing and reduce latency by connecting to computing nodes close to the end users. In the context of sense-process-actuate models, such as those used in healthcare, smart homes, and autonomous vehicles, IoT–fog–cloud systems can ease many parts of our everyday lives.

Novel IoT solutions often rely on trigger-action programming, which can be utilised by smart scene management tools. For instance, IoT applications can benefit from IFTTT [4] trigger-based web services. In an IFTTT scene, various actions and trigger events are linked. For instance, the room temperature can be automatically adjusted based on an IoT sensor measurement. We could call these kinds of combinations of services (e.g., the ones offered by IFTTT) IoT workflows.

Evaluating the effectiveness of novel scheduling and resource provisioning strategies for IoT workflow applications and investigating how they are executed in the fog or in the cloud are currently challenging. These issues are often difficult to test in real environments for the following reasons. In large-scale IoT workflows, it can be difficult to successfully conduct real-world experiments to improve the behaviour of IoT applications, especially when a statistically significant number of experimental results is needed to provide information about possible improvements to IoT applications. For this reason, the scope of research and development in the field of IoT workflow simulation is limited. In the case of experimenting with real IoT systems, researchers are usually able to perform a limited number of different scenarios. Due to the high resource requirements, it is also very difficult and costly to repeat the results of an experiment in a large number of real-world scenarios. As a result, researchers often use simulators. Experimental results can be easily reproduced using a simulation because it provides an environment that is repeatable, controlled, reliable, and scalable. Consequently, a simulator that could support the modelling of IoT workflows would be very useful. It would allow both researchers and industrial parties to model their scenarios and evaluate the performance of their algorithms in a very short time in a realistic simulation environment in a cost-effective manner as well.

Currently, there are few simulation tools and software programs that execute workflows in cloud and fog environments, but the previously specified IoT workflows cannot be defined using the existing generic workflow description standards. To the best of our knowledge, our study is the first step towards making unified tools for describing and simulating IoT workflows.

In this paper, we address the above-mentioned research challenges. We have developed and implemented an IoT workflow extension in the DISSECT-CF-Fog [5] simulator to model IoT workflow applications in fog and cloud environments. It provides the ability to model and simulate the execution of IoT workflows over resources provided by different fog and cloud infrastructures. Our main contributions are as follows: (1) modelling Internet of Things workflow applications; (2) developing a new XML structure for simulating IoT workflow applications; (3) presenting a novel simulator extension that leverages the workflow scheduling and execution capabilities of the DISSECT-CF-Fog simulator to run modelled IoT workflow applications in fog and cloud environments; and (4) evaluating the proposed simulator extension through simulating the IoT and scientific workflows in fog and cloud environments.

The remainder of the work is arranged as follows. Section 2 provides background information and reviews related works on IoT workflow simulators. The details of the design and implementation of our extension are described in Section 3. The results of the experiment are shown and discussed in Section 4. Section 5 concludes this paper and states future work objectives.

## 2. Research Background

Through the spread of distributed and parallel computing, monolithic applications were replaced by modular software, which perfectly fits the architecture of grid computing and cloud computing. In these complex, distributed systems, the logical distribution of the application components is paired with scalable computing resources to facilitate effective execution. These computing-intensive applications typically come from the fields of astronomy, earthquake and ocean science, and bioinformatics, among others. The following challenges have to be taken into account in the case of complex distributed scientific workflow management systems [6]: (1) data movement between and within workflow components in a scalable and parallel manner, (2) resource selection and provisioning in regard to resource types and the costs of data transfers and analysis, and finally (3) effective execution and scheduling of various tasks.

One of the earliest workflow management systems is Pegasus [7], which was initially proposed to execute workflows in grid environments. However, the foundation of Pegasus was laid in 2001, and it can clearly be seen in the literature that it is still an active research area thanks to the unfolding of different computing paradigms such as cloud computing and fog computing. To execute various workflows with Pegasus, they also offered a description format called Directed Acyclic Graph in XML (DAX).

With a directed acyclic graph (DAG), a workflow can be formalised as follows: G=(V,E), where *V* is the set of tasks or jobs (vertices) and *E* is the set of dependencies between the tasks (edges). The number of incoming and outgoing edges (i.e., dependencies) is not restricted to one, except for the start or entry and stop or exit events, where the in and out directions are missing, respectively. In the case of scientific workflows, a dependency typically means a data transfer (e.g., file movement). Therefore the following operations can be represented [8]: (1) processing (one incoming and one outgoing dependency), (2) data distribution (one incoming and multiple outgoing dependencies), (3) data aggregation (multiple incoming and one outgoing dependency), and (4) data redistribution (multiple incoming and multiple outgoing dependencies). Therefore, a task can only be started if its dependencies have been resolved. The edge (i,j)∈E means that the parent task $v_{i}$ has to complete to start the child task $v_{j}$.

In order to execute complex workflows effectively in terms of application makespan, utilisation cost, network utilisation, and energy consumption, workflow scheduling algorithms are applied. However, different scenarios may require different scheduling policies. Finding the optimal scheduling for cloud or fog resources is challenging because (1) the availability of resources cannot be foreseen at the time when a task is ready to be executed, (2) the best fitting resource is hard to find for a task in the case of its migration, (3) the maximal utilisation of the resource-constrained fog environment is difficult to manage in order to achieve the shortest makespan of the workflow application, and finally (4) the execution of the workflow has to meet multi-objective criteria [9].

Since IoT workflows are similar to scientific workflows, mostly in terms of their general representation, execution rules, and research goals, our plan was to first survey the available simulation solutions focusing on workflow modelling. In the following section, we summarise the most significant simulators capable of modelling workflows in simulation environments.

### 2.1. Related Works

WorkflowSim [10] is an extension of the popular CloudSim simulator which aims to model cloud infrastructures and services. It is written in the Java programming language. CloudSim does not support workflow-based simulation by default. Therefore, the extended version provides various modules for workflow management by considering similar components, such as the ones Pegasus WMS has. The most important components are the workflow mapper, which aligns abstract workflows to concrete jobs in the execution environment, the workflow engine, which handles dependencies between the workflow jobs, and the workflow scheduler, which assigns jobs to computing resources. This tool also utilises a clustering engine to resolve scheduling overheads by merging tasks. Since WorkflowSim only supports executing Pegasus trace files by importing DAX files, it is therefore only applicable for executing scientific workflows on cloud resources without the consideration of fog resources and IoT components. Although the simulator is open-source (accessed on GitHub on 30 November 2022: https://github.com/WorkflowSim/WorkflowSim-1.0), it is no longer maintained.

The FogWorkflowSim [11] simulator includes a fog environment for workflow applications to investigate different resource and task management strategies. This tool is built upon iFogSim to support more complex fog topologies as well as WorkflowSim, inheriting the capability of executing workflow applications. Therefore, the WMS of FogWorkflowSim (accessed on GitHub on 30 November 2022: https://github.com/ISEC-AHU/FogWorkflowSim) utilises similar components for transforming the abstract, DAX-based workflow file into tasks executable in the simulator, namely task clustering and scheduling. The authors proposed various greedy and more sophisticated scheduling algorithms such as MaxMin, RoundRobin, and Particle Swarm Optimisation. However, end devices are considered by FogWorkflowSim to be potential computing resources, and unfortunately, IoT-related workflow simulations are totally omitted.

EdgeWorkflow [12] follows a slightly different approach for workflow management. While this engine depends on FogWorkflowSim, it allows the creation of a real computing environment. First, a simulation phase is used to test the theoretical resource and task management strategies (i.e., it basically utilises the components of FogWorkflowSim), to make them comparable with the real execution, and to help decide which offloading and scheduling algorithms should be selected. Second, there is a real phase with a MySQL database and a Docker container pool for executing the workflow in an existing environment by mapping the simulation tasks with real computing tasks. Even if this work aims to simulate edge workflow applications, only scientific workflow descriptions (i.e., DAX) are considered by EdgeWorkflow (accessed on GitHub on 30 November 2022: https://github.com/ISEC-AHU/EdgeWorkflow) for the evaluation.

One of the latest simulators is WIDESim [13], which also extends CloudSim to analyse resource management of workflows in the distributed environments of cloud, edge, and fog computing. The simulator focuses mainly on the network topology of such domains, considering centralised as well as decentralised schemes, and it quits using DAX files. Instead, the authors introduced a JSON-based description for defining a workflow and its dependencies. Unfortunately for the evaluation, only scientific workflows were parsed and compared to concurrent simulators (accessed on GitHub on 5 December 2022: https://github.com/maminrayej/WIDESim).

IoTSim-Stream [14] aims to simulate big data workflow applications with multi-cloud resources. Similar to the previously discussed simulators, this is also a CloudSim-based realisation (accessed on GitHub on 5 December 2022: https://github.com/mutazb999/IoTSim-Stream). However, IoTSim-Stream also deals with a modified DAX-based configuration file and scientific workflows. Still, this work is the closest to ours by considering components as standalone services and incoming data from external sources such as IoT sensors, and the output data can be forwarded in replica or partition mode. Since multiple data flows are considered, therefore, the authors proposed a dedicated data stream scheduling algorithm for stream-based queues of VMs.

In our earlier work, we introduced DISSECT-CF-WMS [15], an extension of DISSECT-CF, which does not support workflow-based simulation. It is designed to perform scientific workflow simulations and analyse internal IaaS behavioural knowledge. It focuses on the user-side behaviour of clouds, while DISSECT-CF focuses on the internal behaviour of IaaS systems. It also supports dynamic provisioning to meet the resource requirements of the workflow application while running on the infrastructure, taking into account the provisioning delay of a cloud-based VM. Since DISSECT-CF-WMS (accessed on GitHub on 4 December 2022: https://github.com/Ali-Alhaboby/Dissect-cf-WMS) only enables the execution of scientific workflow trace files (DAX files), it is only useful for running scientific workflows on cloud resources without considering fog resources or IoT components.

Since none of the simulation tools mentioned above consider the fine-grained IoT functionalities of sensors and actuators or applications utilising fog resources, therefore, our goal is to extend DISSECT-CF-Fog [5] towards the modelling of IoT workflows. The DISSECT-CF-Fog simulator is dedicated to modelling comprehensively the cloud-to-thing continuum [16]. It has a detailed and expansible representation of IoT sensors, actuators, and devices. Furthermore, a multi-layered fog topology can be utilised with realistic network and virtualisation layers, aside from IoT- and cloud-side pricing schemes being based on real providers, and the energy measurements of the nodes are calculated during the simulation. Thus far, the Java-based DISSECT-CF-Fog (accessed on GitHub on 6 December 2022: https://github.com/sed-inf-u-szeged/DISSECT-CF-Fog) only supports batch processing. However, due its flexibility and scalability, it is applicable to modelling stream processing as well. It is also worth mentioning that there are significant performance differences between the core simulators, namely CloudSim and DISSECT-CF. In some cases, DISSECT-CF was 2800 times faster in terms of scalability of the simulation time [17].

A comparison of the discussed simulators can be seen in Table 1. For the comparison, we listed the dependencies of the concrete simulator and the workflow trace support, and we also indicated the year when its source code was modified last. Furthermore, we depicted with *√* the domains which were considered in the simulators.

## 3. The IoT Workflow Simulator

The IoT infrastructure is now being used in a wide range of notable applications, including smart cities, healthcare, and manufacturing [18]. As we highlighted earlier, IoT workflows differ from scientific workflows as additional aspects need to be considered. This kind of workflow follows a trigger-action programming model [4] typically offered by a third-party service provider such as Google Assistant. In such cases, the focus may be shifted from the total execution time to the trigger-to-action (T2A) latency, considering it in various scenarios.

Due to the financial implications and the need for broad-based behaviour, complex IoT workflows cannot be explored on a large scale when the number of entities involved reaches thousands in real life. To simulate independent IoT workflows in addition to generic scientific workflows, we developed a simulation extension of DISSECT-CF-Fog focusing on IoT devices and applications with realistic network settings, which can be configured according to the T2A values.

In [19], trigger- and action-based workflow management systems are presented with real evaluation of IoT scenarios. Through using and extending this, and to further illustrate the differences and similarities, let us depict a normal IoT process in the office, as shown in Figure 1. There are still communication and data dependencies between jobs (e.g., the impossibility to start processing data until every sensor has sent a data packet and it has arrived at the computing node). As a result, computing tasks typically utilising VMs are still present in the system, similar to scientific workflows. Time-dependent triggering events are a new addition to the system. For example, when a user (in this case a programmer) enters the office, a service triggers some kind of event by sending an instruction to an actuator to execute it (e.g., turning on or off the air conditioning, lighting, or coffee machine). In addition, the system contains time-dependent repetitive events, such as measuring the temperature and humidity in the office every 60 s until the programmer leaves the office. The goal of the IoT workflow simulator is to optimise the performance of IoT applications as a whole, as well as the performance of their individual components.

To perform such events in the DISSECT-CF-Fog simulator, the former actuator model of the simulator had to be slightly modified. In this work, the actuator representation can consider the speed of action. Actuators are active components that control the system or a mechanism utilising a network connection as well. In a simulation environment, such control (e.g., moving a robot arm using some kind of energy source, such as a hydraulic system) cannot be perfectly modelled due to the limitation of the abstraction, and thus we only considered the necessary time to perform such an action (T2A). This means that if more physical actuators are present in the simulation with different speeds of action, then greater importance is attached to the scheduling of actuator tasks. The sensor model, which is a passive entity, remains as it was before in the simulator. When a sensor event is triggered or activated, data obtained via a measurement with the predefined size are stored locally in the IoT device, to which the sensor is connected. Based on the network characteristics of the IoT device (i.e., bandwidth and latency), the data are forwarded to a computing node, where the corresponding service is assigned.

The initial plans include the implementation of three more modules in DISSECT-CF-Fog: (1) an initialisation unit that is able to convert abstract workflows into simulated tasks, (2) an execution engine built upon the discrete event system of the simulator, which assigns tasks to the appropriate executors depending on the type of task, and (3) monitoring and reporting procedures that analyse the results. The flow chart of the extended DISSECT-CF-Fog towards IoT workflow modelling can be seen in Figure 2, which summarises the whole process in 13 steps.


**Phase I. Initialisation**
−*Step 1. Generating and loading DAX*: In this step, an input file containing the description of an abstract workflow is loaded. The IoT workflow generator module of the simulator can also be used for defining the required number and type of jobs.−*Step 2. Converting to IoT format*: If the input workflow file contains the description of the scientific workflow (e.g., Epigenomics), then the converter module of the simulator is applied to parse it to the proposed new format.−*Step 3. Creating jobs*: Only the new IoT-based description is accepted by the simulator, while the abstract jobs defined are transformed into Java objects in this step.−*Step 4. Visualising graph*: The only output of the first phase is a DAX visualiser, which shows the jobs, their dependencies, and the complexity of the graph.−*Step 5. Submitting workflow scheduler*: The goal of the scheduler is to define the pre-simulation operations (e.g., defining a comparator procedure to order the ready jobs in a queue), job allocation policies (e.g., to manage file transfers, an initial storage location has to be associated with the jobs), and resource scaling decisions (increasing or reducing VMs).−*Step 6. Loading the physical topology*: The physical topology containing cloud, fog and IoT resources with position, capacity, and network properties must be configured initially.
**Phase II. Execution**
−*Step 7. Submitting jobs to the executor engine*: The workflow engine manages the events and observes the workflow scheduler. It also determines the entry jobs having no dependency.−*Step 8. Initialising VMs and job queues for resources*: The simulation can be started only when the initialisation jobs of the scheduler are executed (e.g., there is at least one running VM for each computing resource) and the job priority for the queues is determined. This step largely depends on Step 5.−*Step 9. Assigning jobs to resources*: A preliminary allocation of jobs to resources is executed in order to carry out various events. This is typically related to the file transfers, which can represent dependencies among services. In DISSECT-CF-Fog, the file transfers between the repositories (i.e., storage) of physical machines, which are equipped with network properties (latency and bandwidth). Therefore, the job assignment for computing nodes at the zero moment of the simulation is inevitable in order to fulfil the dependencies whenever a job is completed.The allocation of the task is based on the scheduling algorithm proposed in Step 5. The initial allocation has a distinguished role in order to minimise the time spent on file transfers. It is feasible if the dependent tasks are allocated to adjacent (or the same) nodes. However, a less even distribution of jobs on nodes may increase the execution time of the workflow.Therefore, various parameters of the system can be involved to create an optimal scheduling of jobs, such as the topology of nodes, network parameters, the strength of the computing environment, and the current workload of nodes.−*Step 10. Polling the queues for jobs and processing them on the proper resources*: Until there are unprocessed jobs, the queues are polled continuously.−*Step 11. When a job is complete, reassigning jobs to resources*: When a job is complete, a reallocation function is called to optimise the utilisation of the resources by reorganising the rest of the jobs. If a service receives some files and then is reallocated to another computing resource, then this also means a further data transfer of those files from the old storage to the new one. Similar to Step 9, different parameters can be considered to improve the system performance in workflow execution.
**Phase III. Reporting**
−*Step 12. Printing information*: When the execution of the jobs is finished, the most valuable information about the simulation is written to the output, such as the execution time, energy consumption, utilisation costs, utilised VMs, and number of jobs reassigned.−*Step 13. Visualising scheduling timeline*: Another visualiser helps with overviewing the scheduling of jobs on resources over time.

As we can see in Section 2.1, most of the concurrent simulators use the DAX format in order to define jobs and dependencies. Furthermore, this schema is well known and commonly used among researchers as well. Since the original version was created to execute real workflows, for simulation purposes, the description cannot be considered transparent and human-readable. In a simulation environment, we dispense with certain addictions, but the simulation still remains realistic. For instance, even if a job depends on many input files, simulators only consider an abstract representation of the files.

As can be seen in the following listing (Listing 1), the original DAX method has two main parts. First, the jobs are defined with *job* tags (Lines 2–6), which is then followed by the definition of the dependencies with *child* and *parent* XML tags (Lines 7–10). If the number of jobs exceeds thousands, then the length of the file may become extremely long with unmanageable dependencies.

**Listing 1.** The original DAX method.

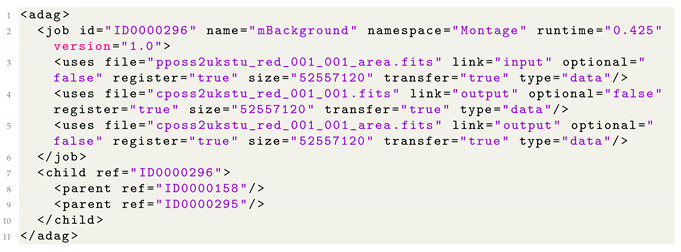



To improve the readability and ease the definition of IoT workflows, we introduce our own DAX-based structure to define IoT workflows in DISSECT-CF-Fog, as is depicted in the following listing (Listing 2). First, we outsourced the definition of the dependencies into the jobs. Therefore, in the case of the interpretation of a job, we can clearly see the incoming and outgoing edges, and thus the *child* and *parent* tags are no longer needed. We also simplified and removed the unused XML tags and attributes to increase transparency.

**Listing 2.** Our own DAX-based structure to define IoT workflows in DISSECT-CF-Fog.

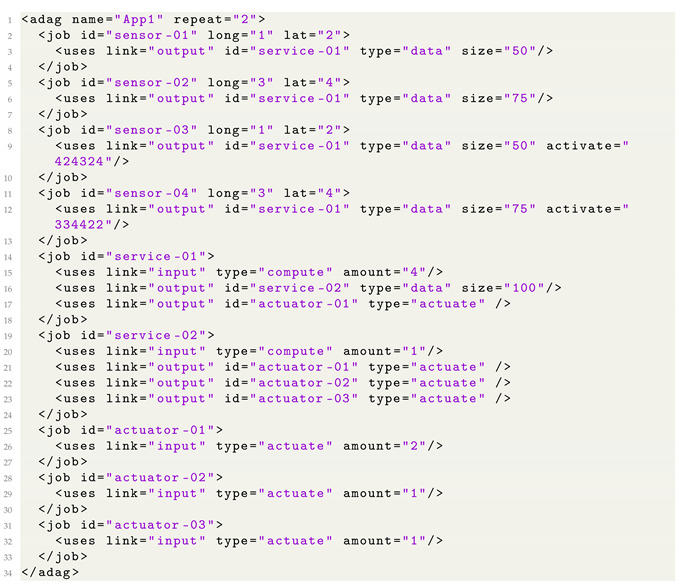



The *repeat* tag promotes the scalability of the defined workflow by copying the jobs individually. The *id* attribute is the unique identifier of each job (defined with *job* XML tags). Furthermore, three types of jobs are distinguished currently: sensor, service, and actuator. Only the sensor type among these represents physical devices, and therefore the *long* and *lat* values show the geographical positions of these kinds of devices (Lines 2–4). Meanwhile, the type of service and actuator are logical components. The reason for this decision is that in an IoT use case, the number of sensors can exceed hundreds of thousands, which makes the program-level definition of those difficult. In the case of service jobs, those are executed with computing resources (i.e., fog and cloud nodes), while in the case of actuator jobs, the actuator entities (i.e., robot arm or coffee machine) are responsible for executing the actualisation event. This means that service and actuator jobs can be reassigned among the existing executors.

For defining the incoming and outgoing dependencies of a job, we have to use the *uses* tags. The *link* attribute defines its direction (i.e., input or output). If it is an output dependency, then we have to use the *type* attribute as well as denote whether it is a data transfer or an actualisation task. If it is the former, then we also have to apply the *size* attribute to define the amount of bytes that should be transferred. There is a dedicated attribute called *activate*, which can be used to define deferred events. For instance, it can define a human interaction or a sensor measurement a certain time later (Lines 16 and 17). Otherwise, it is considered to be zero, meaning there is an immediate execution right after the simulation starts. Finally, if an input dependency is defined for a job, then its quantity should be defined using the *amount* attribute (Line 20). As we mentioned earlier, we merged the definition of jobs and dependencies, and therefore each *uses* tag deals with an *id* tag denoting the identifier of the dependent job.

## 4. Evaluation

In our evaluations, we used the model of a Hungarian private cloud infrastructure, namely the ELKH Cloud [20], and we assumed the following use cases. First, a representative IoT use case is presented, introducing the possible events proposed in this paper, and then a scalability study is executed. Finally, we show the interoperability of the system, concerning the execution of traditional scientific workflows. In this paper, our goal was not to investigate different scheduling algorithms but to show a novel way of modelling IoT systems and to exemplify the usability of DISSECT-CF-Fog. Nevertheless, we needed to implement a scheduling algorithm considering each type of job, and therefore we used a traditional scheduling algorithm that mapped service tasks to fog, cloud, and actuator resources. In the case of computing resources, it sorts the tasks in the queue in descending order according to the length of their runtimes. Finally, the task with the longest runtime is scheduled on the fastest or strongest available resource. In the case of actuators, the jobs are assigned following first-come-first-served (FCFS) scheduling.

In order to construct a multi-layered fog-cloud topology, we applied our methodology as follows. For each cloud node, we set 2–4 fog nodes (depending on the number of fog and cloud nodes predefined in the concrete scenarios) to form a cluster together. Since DISSECT-CF-Fog can take into account the positions of the nodes, and the latency values (50 ms, the average 4G delay) between the nodes are weighted with the physical distance, the final latency was thus between 100 and 130 ms. In the case of cloud nodes, their bandwidth was set to 1600 Mbps, whilst for the fog nodes, it was reduced to 1000 Mbps. The further parameters (CPU cores, memory, etc.) of the computing nodes can be seen in Table 2, which were used in all scenarios.

### 4.1. Morning Routine Scenario

In this IoT use case, we would like to show the broad functionality of our proposed simulator extension. Therefore, we first realised the scenario presented in Figure 1. Since that figure is just an abstract representation, we thus considered it with the following realisation. The workflow file contains five programmers checking in in the first minute. Four out of five programmers intend to start the day with a cup of coffee, which means four triggering events. Currently, the office has only two coffee machines: an old slower one and a fast new one (i.e., in the case of scheduling the actuator events, it matters which job is assigned to which machine). There is another actuator event parallel with the coffee making which turns the lights on in the office rooms. In this case, this event is considered to be executed with a switch, and therefore these kinds of events only utilise the network.

In a smart office, the air quality can also be monitored by different sensors. Thus, in this case, a sensor samples the environment every minute for 20 min. The data are forwarded to a service, and to be as realistic as possible, in three cases, low temperatures are detected. Therefore, in such cases, these service events trigger the heating system (i.e., actuators). Similar to the light actuators, these instructions only utilise the network connection. Finally, after 20 min, the programmers check out, resulting in five further actuator events.

Since the morning routine scenario describes the first 20 min of five programmers’ days, we sought the necessary amount of cloud, fog, and actuator resources to keep the time around 20 min while the number of programmers increased. Therefore, in the first case, we repeated the jobs once (i.e., five programmers), and in the second case, we set the repeats to 10, which meant 50 programmers. In the last use case, we evaluated the workflow with 100 repetitions (i.e., a business with 500 programmers). The results can be seen in Table 3 and Table 4. As one can see, a single cloud was enough to provide the minimum computing and actuating capacity for five programmers, but as the number of programmers and thus the data increased, both the execution cost and the energy consumption dramatically increased. We needed 100 cloud and 400 fog nodes as well as 100 actuators to keep the execution time as close to 20 min as possible. It depicted the order of magnitude of the necessary physical resources.

### 4.2. Performance and Scalability of IoT Workflows

We investigated the performance of IoT workflows in terms of execution time, CPU utilisation, energy consumption, and total amount of data processed in applications ranging from very small to very large. This experiment demonstrates the simulator’s ability to model, simulate, and schedule both simple and sophisticated IoT workflow applications in fog and cloud environments. This enables researchers to investigate the behaviour of IoT workflow application topologies and configuration sizes for future evaluation and development, such as by developing new deployment and scheduling strategies, improving execution performance, and investigating QoS and SLA requirements for these types of applications.

The goal of the initial experiments was to evaluate the performance and scalability of the IoT workflow simulator with different configuration sizes for workflow applications. We conducted these experiments using a laptop equipped with an Intel Core i7-8750H CPU @ 2.20 GHz (with 6 cores and 12 logical processors), 16GB RAM, and Windows 10 Enterprise and then collected the experimental results. All experiments were evaluated with synthetic IoT workflows derived from our IoT generator.

Table 5 shows the total IoT workflow entities of the sensors, services, and actuators in each workflow size for the application. We increased the total number of entities in each experiment by 10,000 to test the scalability up to a simulation time of 10 min. Table 6 illustrates the number of resources used in the evaluation.

We collected performance metrics for the experiments until we reached 440,000 entities executed. Table 7 shows the collected metrics for each experiment with DISSECT-CF-Fog on fog and cloud resources. The execution time for simulating IoT workflows increased by an average of 1.21 times from 10,000 to 110,000 in configuration size for all simulated workflow applications. From 110,000 to 440,000, this rate increased to 3.45 times, showing that the physical infrastructure became more overloaded, reaching 95.75% utilisation.

With a total of 10,000 entities, only slightly less than 78% of the rented resources were required by the fog and cloud resources. This pattern repeated (with almost gradual increases) for all other IoT workflows. The IoT workflows had many parallel service-type tasks with a short execution time in the second layer of the structure, as shown in Figure 3. This led to a drastic increase in the overall cost of the workflow, as more resources were consumed and the resource utilisation of the VMs increased.

If we compare the experiments in terms of energy consumption and processed data, the results show that they increased as the number of entities increased. The sensor jobs sent a large amount of data to the service jobs to process. Note that the increase in processed data and energy consumption was significant when the number of entities increased.

This behaviour for the simulation time was to be expected, as the number of entities increased 10-fold for a large configuration and also significantly increased the total execution time of the IoT workflows executed by these applications. The execution time for the simulation of the IoT workflow with 100,000 entities was less than 8 min, while the simulation of the IoT workflow with 110,000 entities took over 11 min. The average increase in the simulation time was only 1.95 times. Thus, DISSECT-CF-Fog was able to analyse the behaviour of different IoT workflow execution scenarios with certain configuration sizes to perform further evaluations and improvements. This could lead to the development of new deployment and scheduling strategies, improved execution performance, and investigating QoS and SLA requirements for IoT workflow applications.

### 4.3. Scientific Workflows

In this subsection, all experiments were evaluated with scientific workflows derived from the Montage (astronomy), CyberShake (earthquake science), LIGO (gravitational physics), and SIPHT (biology) applications, taking into account data transfers. The Montage [21] workflow is an astronomy application used to generate custom mosaics of the sky based on a set of input images. The CyberShake [22] workflow is used to characterise earthquake hazards by generating synthetic seismograms. The Laser Interferometer Gravitational Wave Observatory (LIGO) [23] workflow is used to analyse data from the coalescing of compact binary systems, such as binary neutron stars and black holes. The sRNA Identification Protocol using High-throughput Technology (SIPHT) programme [24] uses a workflow to automate the search for sRNA-encoding genes for all bacterial replicons in the National Center for Biotechnology Information (NCBI) database. Figure 4 presents the structures of the four workflows.

We collected the same performance metrics that were collected for the IoT workflow application. Again, we used the same number and configuration of fog and cloud resources mentioned in Section 4.2. The scientific workflows were run on fog and cloud resources. Table 8 shows the experimental results of the scientific workflows. This experiment illustrated that the simulator could model, simulate, and schedule scientific workflows in both fog and cloud environments. The number of tasks for each workflow was 1,000 tasks used for the evaluation.

In the utilisation of VM resources, the maximum percentage was below 60% because the number of VM resources was greater than the maximum number of parallel tasks for all scientific workflow applications. Table 8 also shows that LIGO had the best average VM utilisation across all applications. This is because LIGO is a data- and CPU-intensive workflow, which significantly increases the utilisation of fog and cloud resources. However, Sipht had the lowest average virtual machine usage among all applications. Some tasks in Sipht had significant variations in their execution times, with a maximum of a 20-fold difference in execution time. This created idle time for other resources and scheduling gaps between workflow tasks, resulting in the lowest resource utilisation. In addition, Sipht had the longest execution time among all applications, resulting in the highest execution cost of about USD 95. In the case of the Montage workflow, Montage consists of several simultaneous tasks that can be completed in a very short time. For this reason, the resource utilisation of the workflow was slightly more than 20% as more resources were used. In the case of the CyberShake workflow, CyberShake had the largest amount of processed data across all applications at 265.89 GB, as it is the bottleneck in data transfer for most scheduling algorithms.

Finally, we turned our attention to demonstrating the benefits of using the IoT and scientific workflows in DISSECT-CF-Fog. This allows developers to evaluate the performance of IoT and scientific workflow applications in fog and cloud environments and discuss the collected results. This allows researchers to study the behaviour of IoT and scientific workflow applications, to design new scheduling algorithms, to improve execution performance, and to evaluate QoS and SLA requirements.

## 5. Conclusions and Future Works

In this paper, we presented the development and validation of a state-of-the-art simulation solution to model IoT workflows, built upon the fundamentals of the DISSECT-CF-Fog simulator. Since the cloud-to-thing continuum is continuously growing, novel solutions are required to fill research gaps in this field. To the best of our knowledge, this simulation approach is currently the first of its kind, targeting the analysis of IoT-Fog-Cloud applications modelled as IoT workflows composed of trigger- and action-based IoT events.

To show the usability and performance of our proposed software, we first presented a typical IoT use case of a morning routine in a software development company, and then we showed a scalability study, where the number of sensor, actuator, and service events was increased to hundreds of thousands. Finally, to compare our approach to scientific workflows and exemplify its generality, we evaluated the simulator with well-known traditional workflows executed in a fog-enhanced cloud environment.

In our future work, we plan to extend the available functionalities of our simulator to be able model more specific, specialised IoT use cases, such as healthcare and smart transportation. We will also aim to investigate different workflow scheduling algorithms tailored to IoT use cases and compare DISSECT-CF-Fog’s performance with other simulation solutions.

## Figures and Tables

**Figure 1 sensors-23-01294-f001:**
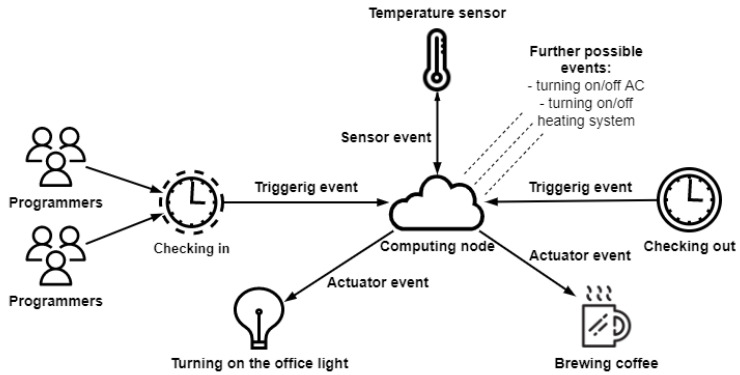
Office routine workflow.

**Figure 2 sensors-23-01294-f002:**
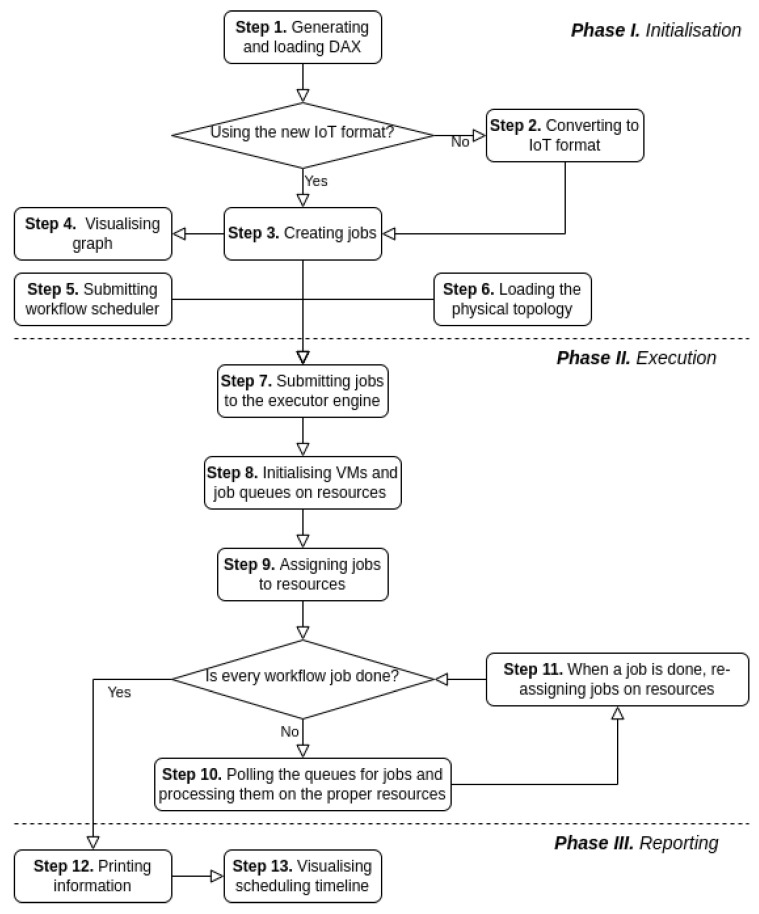
Flow chart of the extended simulator towards IoT workflow modelling.

**Figure 3 sensors-23-01294-f003:**
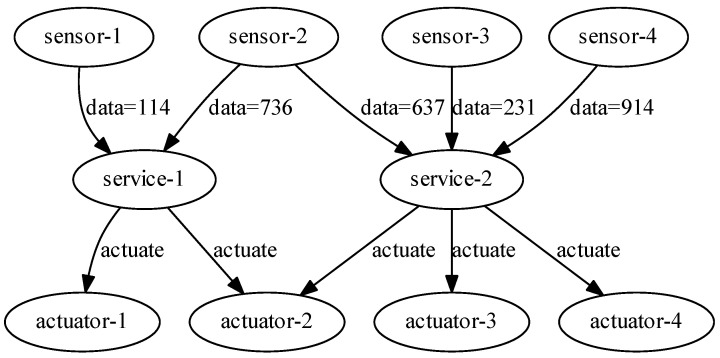
A sample IoT workflow with sensors, services, and actuators.

**Figure 4 sensors-23-01294-f004:**
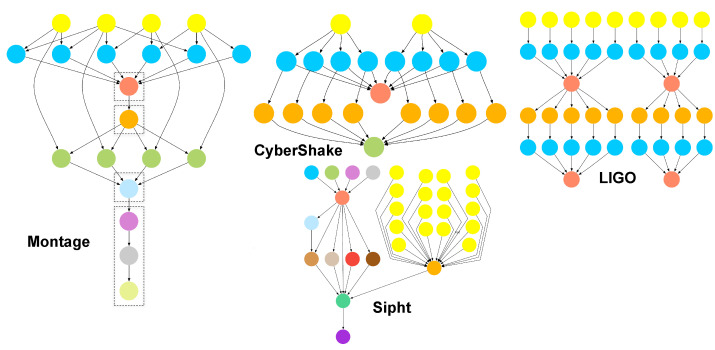
The structures of the Montage, CyberShake, LIGO, and Sipht workflows.

**Table 1 sensors-23-01294-t001:** Comparison of the related workflow simulators.

Simulator	Last Modified	Dependency	Trace Support	IoT	Domains of Fog or Edge	Cloud
WorkflowSim	2015	CloudSim	DAX	-	-	*√*
FogWorkflowSim	2019	iFogSim, WorkflowSim		-	*√*	*√*
EdgeWorkflow	2021	FogWorkflowSim		-	*√*	*√*
IoTSim-Stream	2021	CloudSim		*√*	-	*√*
WIDESim	2022	CloudSim	JSON	-	*√*	*√*
DISSECT-CF-WMS	2022	DISSECT-CF	DAX	-	-	*√*
DISSECT-CF-Fog	2022	DISSECT-CF		*√*	*√*	*√*

**Table 2 sensors-23-01294-t002:** Configuration parameters and cost of resources used in the evaluation experiments.

Configuration Parameters	Fog Resources (PM)	Fog Resources (VM)	Cloud Resources (VM)	Cloud Resources (PM)
CPU (Cores)	4	2	4	64
CPU (MIPS)	1000	1000	1000	1000
RAM (GB)	4	2	4	256
Storage (GB)	512	256	256	500
Price (hourly)	-	0.051	0.102	-

**Table 3 sensors-23-01294-t003:** Requirement of resources used in the evaluation experiments.

Workflow Repeat	Cloud Resources (VM with 8 Cores)	Fog Resources VM with 4 Cores)	Actuator Resources
1	1	-	12
10	12	10	30
100	100	400	100

**Table 4 sensors-23-01294-t004:** The experimental results of DISSECT-CF-Fog with IoT workflow of morning routine scenario in Figure 1.

Workflow Repeat	Execution Time (min)	Simulation Time (ms)	VMs Utilisation (%)	Execution Cost ($)	Processed Data (MB)	Energy Consumption (kWh)
1	21	567	53.36	0.204	10.5	2.77
10	22	1012	74.45	3.468	106	5.9
100	23	2667	89.59	61.2	1075	74.94

**Table 5 sensors-23-01294-t005:** The entity number of sensors, services, and actuators in each workflow determines the size of the application.

Workflow ID	Sensor Events	Service Events	Actuator Events	Total Entities
1	3400	3300	3,300	10,000
2	7000	6500	6,500	20,000
3	10,000	10,000	10,000	30,000
4	14,000	13,000	13,000	40,000
5	20,000	15,000	15,000	50,000
6	20,000	20,000	20,000	60,000
7	24,000	23,000	23,000	70,000
8	28,000	26,000	26,000	80,000
9	30,000	30,000	30,000	90,000
10	34,000	33,000	33,000	100,000
11	40,000	35,000	35,000	110,000
12	80,000	70,000	70,000	220,000
13	160,000	140,000	140,000	440,000

**Table 6 sensors-23-01294-t006:** Requirement of resources used in the evaluation experiments.

Parameter	Value
Fog resources (PMs)	100
Fog resources (VMs)	200
Cloud resources (PMs)	12
Cloud resources (VMs)	200
Actuator resources	200
Clouds	2

**Table 7 sensors-23-01294-t007:** The experimental results of DISSECT-CF-Fog with different numbers of IoT workflows.

id	Execution Time (min)	Simulation Time (s)	VMs Utilisation (%)	Execution Cost ($)	Processed Data (GB)	Energy Consumption (kWh)
1	33	4.537	77.31	39.63	1.69	26.79
2	53	21.820	82.9	157	3.5	44.32
3	66	40.228	86.06	350	5	57.26
4	88	49.706	89.16	616	7	76.9
5	110	88.648	87.51	961	10	98.9
6	122	127.704	89.93	1,374	10.5	106.81
7	141	265.071	88.93	1,853	11.9	124.4
8	156	276.995	90.13	2,435	14	140.9
9	172	334.261	90.39	3,065	15	153.99
10	189	459.554	90.82	3,793	17	169.49
11	209	690	92.46	4,605	20	189.76
12	957	1878	94.23	62071	40	928.8
13	2240	4495	95.78	248210	79.98	2043

**Table 8 sensors-23-01294-t008:** The experimental results for DISSECT-CF-Fog with different scientific workflow applications.

Workflow	Execution Time (min)	Simulation Time (s)	VMs Utilisation (%)	Execution Cost ($)	Processed Data (GB)	Energy Consumption (kWh)
Montage	17.5	1.283	30.38	16.45	14.45	12.27
CyberShake	20.8	1.144	35.94	23.5	265.89	14.83
LIGO	52.3	1.01	59.65	45.4	0.39	41.42
Sipht	124.6	1.09	21.68	94.7	2.88	88.38

## Data Availability

The source code of DISSECT-CF-Fog is available on GitHub https://github.com/sed-inf-u-szeged/DISSECT-CF-Fog (accessed on 6 December 2022).

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
