# Peer review of "Simulating IoT Workflows in DISSECT-CF-Fog"

_sensors, 2023, doi:10.3390/s23031294_

Round 1

Reviewer 1 Report

The background that led to the Importance is clear. This paper presents interesting results on Simulating IoT Workflows in DISSECT-CF-Fog. The quality of the research work presented in the paper is manageable. This paper is properly directed but  some grammatical/typological errors must be corrected specially in the introduction part.

Author Response

The background that led to the Importance is clear. This paper presents interesting results on Simulating IoT Workflows in DISSECT-CF-Fog. The quality of the research work presented in the paper is manageable. This paper is properly directed but  some grammatical/typological errors must be corrected specially in the introduction part.

Answer: Thank you very much for your time and the valuable comments on the manuscript. We proofread the text to improve the grammar.

Reviewer 2 Report

The article fairly well-written and addresses an important problem in the domain of IoT simulations.

The proposed IoT simulation framework will greatly help the scientific community in evaluation and simulating IoT workflows.

I have following comments/suggestions:

1) The evaluate the scalability of proposed simulator more experiments with larger synthetic workflows may be conducted.

2) Performance comparison with most closely related simulator, for instance, IoTSim-Stream may be included.

Author Response

The article fairly well-written and addresses an important problem in the domain of IoT simulations. The proposed IoT simulation framework will greatly help the scientific community in evaluation and simulating IoT workflows.

Answer: Thank you very much for your time and the valuable comments on the manuscript. We have made a revision of the paper regarding the changes requested, and we tried to take all your comments into account. The detailed response to the comments follows below.

The evaluate the scalability of proposed simulator more experiments with larger synthetic workflows may be conducted.

Answer: Thank you for your suggestion. In the previous version, our criteria for the scalability was to increase the number of entities until the simulation time stays below 10 minutes.  In the revision, we added two extra measurements where we doubled twice the number of entities to show the performance of the simulator on larger workflows as well. 

In Section 4.2, the text and the related tables were modified accordingly.

Performance comparison with most closely related simulator, for instance, IoTSim-Stream may be included.

Answer: We definitely agree that such comparison would be beneficial, but to the best of our knowledge, there is no other simulation solution involving similar IoT workflow and utilising IoT-Fog-Cloud layers as well. Currently, the comparison can be evaluated only with scientific/classical workflows, which is not the main contribution of our paper. Furthermore we feel that such performance comparison would require a whole paper focusing on the different components of the simulators, performance and exhausting evaluation of scientific workflows, therefore we added this kind of evaluation as a future work.

Reviewer 3 Report

In this paper, the authors proposed a simulator extension of the DISSECT-CF-Fog simulator. 

a) Introduction part is good, authors provided research issues and their contribution in Introduction.

b) Related work is good, but please include details about open source or cost details of each simulator.

c) It would be good, if authors compares DISSECT-CF-Fog with other simulators.  

d) Please check  typo / language mistakes before publications.

Author Response

In this paper, the authors proposed a simulator extension of the DISSECT-CF-Fog simulator. Introduction part is good, authors provided research issues and their contribution in Introduction.

Answer: Thank you very much for your time and the valuable comments on the manuscript. We have made a revision of the paper regarding the changes requested, and we tried to take all your comments into account. The detailed response to the comments follows below.

Related work is good, but please include details about open source or cost details of each simulator.

Answer: The GitHub availability of the simulators is placed in footnotes, all of them are open-source and can be utilised without any extra fees. 

It would be good, if authors compares DISSECT-CF-Fog with other simulators.  

Answer: We definitely agree that such comparison would be beneficial, but to the best of our knowledge, there is no other simulation solution involving similar IoT workflow and utilising IoT-Fog-Cloud layers as well. Currently, the comparison can be evaluated only with scientific/classical workflows, which is not the main contribution of our paper. Furthermore we feel that such performance comparison would require a whole paper focusing on the different components of the simulators, performance and exhausting evaluation of scientific workflows, therefore we added this kind of evaluation as a future work.

Please check  typo / language mistakes before publications.

Answer: We proofread the text to improve the grammar.

Reviewer 4 Report

In this paper, authors propose a simulator for fog computing applications.
The simulator is an extension of DISSECT-CF to support direct acyclic graph to describe the workflow of computations.

The papers presents well the work accomplished even if the theoretical part is small in this kind of papers.

Here are few comments for each part.

Related works:

- What about ifgosim (https://github.com/Cloudslab/iFogSim) that have been used in many papers on fog computing and that have thought to be the reference simulator for fog applications.

The IOT Worflow simulator

- Why modifying DISSECT-CF and not the previous presented solutions. For instance why not adding a module to transform DAX to JSON in WIDESIm? Or modifying how iot devices are managed in FogWorkflowsim? Is it because many solutions depend on DISSECT-CF and the solution can be therefore more easily integrated in other products?

  - how the simulator deals with network topology and mobility of nodes. In my mind it is not sufficient to describe the workflow of computations executed on different nodes but the latency/throughput/node selection rules are also important. For instance, a user may wants its iot node to connect the fog node with the lowest latency to perform a computation but if this node is saturated or if it does not have the sufficient resources (cpu/ram/disk) the iot node can send its computation to another node.

In the same way, how the simulator deals with mobility of nodes (to simulate a moving vehicle that connects the closest fog site for instance)

I think the step 9 of the Figure2 requires more detailed due to the fog computing context.

Evaluation

I am happy to see that the simulator was tested on different classical workflows. It should be great addition to the paper if we have a comparison with the performance of other simulators.

Author Response

In this paper, authors propose a simulator for fog computing applications. The simulator is an extension of DISSECT-CF to support direct acyclic graph to describe the workflow of computations. The papers presents well the work accomplished even if the theoretical part is small in this kind of papers.

Answer: Thank you very much for your time and the valuable comments on the manuscript. We have made a revision of the paper regarding the changes requested, and we tried to take all your comments into account. The detailed response to the comments follows below.

What about ifgosim (https://github.com/Cloudslab/iFogSim) that have been used in many papers on fog computing and that have thought to be the reference simulator for fog applications.

Answer: iFogSim has had a presence in the field for a long time, but to the best of our knowledge, it does not support (IoT) workflow simulation by default. However, iFogSim-based FogWorkflow is considered in the paper. 

Why modifying DISSECT-CF and not the previous presented solutions. For instance why not adding a module to transform DAX to JSON in WIDESIm? Or modifying how iot devices are managed in FogWorkflowsim? Is it because many solutions depend on DISSECT-CF and the solution can be therefore more easily integrated in other products?

Answer: The reason why we chose DISSECT-CF was twofold. DISSECT-CF is proven to be occasionally 2,800-times faster than CloudSim [1], this tendency is also observable between the derived DISSECT-CF-Fog and iFogSim [2], showing 10-times performance difference in favour of DISSECT-CF-Fog. Furthermore, the previous version of DISSECT-CF-Fog has already supported batch processing of IoT data, which facilitated the integration of stream processing of workflows.

[1] Z. Mann. Cloud simulators in the implementation and evaluation of virtual machine placement algorithms. Software: Practice and Experience, 48:7, 2017. https://doi.org/10.1002/spe.2579.

[2] A. Markus and A. Kertesz. Investigating IoT Application Behaviour in Simulated Fog Environments. Cloud Computing and Services Science, 258–276, 2021. https://doi.org/10.1007/978-3-030-72369-9_11

How the simulator deals with network topology and mobility of nodes. In my mind it is not sufficient to describe the workflow of computations executed on different nodes but the latency/throughput/node selection rules are also important. For instance, a user may wants its iot node to connect the fog node with the lowest latency to perform a computation but if this node is saturated or if it does not have the sufficient resources (cpu/ram/disk) the iot node can send its computation to another node.

Answer: The DISSECT-CF-Fog has a realistic network model including latency and bandwidth parameters, which is taken into consideration in the evaluation for the multi-layered fog-cloud topology. However, thank you for the suggestion, in the future we will consider the network properties for a more sophisticated scheduling algorithm as well. 

We extended the evaluation with a more detailed description of the network parameters in Section 4.

In the same way, how the simulator deals with mobility of nodes (to simulate a moving vehicle that connects the closest fog site for instance)

Answer: The DISSECT-CF-Fog is able to model various movements of IoT devices [1], however it was not considered in the evaluation, since it also requires the improvements of the scheduling algorithm. 

[1] A. Markus, M. Biro and A. Kertesz. Actuator behaviour modelling in IoT-Fog-Cloud simulation. 7(5):e651, 2021. https://doi.org/10.7717/peerj-cs.651

I think the step 9 of the Figure2 requires more detailed due to the fog computing context.

 Answer: We agree with the reviewer, thank you for your  suggestion. We modified the description of Step9 in the text accordingly.

I am happy to see that the simulator was tested on different classical workflows. It should be great addition to the paper if we have a comparison with the performance of other simulators.

Answer: We definitely agree that such comparison would be beneficial, but to the best of our knowledge, there is no other simulation solution involving similar IoT workflow and utilising IoT-Fog-Cloud layers as well. Currently, the comparison can be evaluated only with scientific/classical workflows, which is not the main contribution of our paper. Furthermore we feel that such performance comparison would require a whole paper focusing on the different components of the simulators, performance and exhausting evaluation of scientific workflows, therefore we added this kind of evaluation as a future work.